# Design, Synthesis and Antibacterial Evaluation of 3-Substituted Ocotillol-Type Derivatives

**DOI:** 10.3390/molecules23123320

**Published:** 2018-12-14

**Authors:** Kai-Yi Wang, Zhi-Wen Zhou, Heng-Yuan Zhang, Yu-Cheng Cao, Jin-Yi Xu, Cong Ma, Qing-Guo Meng, Yi Bi

**Affiliations:** 1School of Pharmacy, Key Laboratory of Molecular Pharmacology and Drug Evaluation, Ministry of Education, Collaborative Innovation Center of Advanced Drug Delivery System and Biotech Drugs in Universities of Shandong, Yantai University, Yantai 264005, China; wangky_1994@163.com (K.-Y.W.); cchm1187829@163.com (Y.-C.C.); 2State Key Laboratory of Natural Medicines and Department of Medicinal Chemistry, China Pharmaceutical University, Nanjing 210009, China; 18694088106@163.com (Z.-W.Z.); panacea0928@163.com (H.-Y.Z.); jinyixu@china.com (J.-Y.X.); 3Department of Applied Biology and Chemical Technology, and State Key Laboratory of Chirosciences, The Hong Kong Polytechnic University, Hung Hom, Kowloon, Hong Kong, China; cong.ma@polyu.edu.hk

**Keywords:** triterpenoid saponin, ocotillol, antibacterial activity, synthesis, structure-activity relationship

## Abstract

Antibiotic resistance has become a serious global problem that threatens public health. In our previous work, we found that ocotillol-type triterpenoid saponin showed good antibacterial activity. Based on preliminary structure-activity relationship, novel serious C-3 substituted ocotillol-type derivatives **7**–**26** were designed and synthesized. The in vitro antibacterial activity was tested on five bacterial strains (*B. subtilis* 168, *S. aureus* RN4220, *E. coli* DH5α, *A. baum* ATCC19606 and MRSA USA300) and compared with the tests on contrast. Among these derivatives, C-3 position free hydroxyl substituted compounds **7**–**14**, showed good antibacterial activity against Gram-positive bacteria. Furthermore, compound **22** exhibited excellent antibacterial activity with minimum inhibitory concentrations (MIC) values of 2 μg/mL against MRSA USA300 and 4 μg/mL against *B. subtilis*. The structure-activity relationships of all current ocotillol-type derivatives our team synthesised were summarized. In addition, the prediction of absorption, distribution, metabolism, and excretion (ADME) properties and the study of pharmacophores were also conducted. These results can provide a guide to further design and synthesis works.

## 1. Introduction

Since the discovery of penicillin, antimicrobial drugs have saved millions of people’s lives and eased patients suffering from chronic infections. However, the emergence of antibiotic resistance has become a serious global problem that threatens public health, and the development of novel treatments and agents is extremely urgent [1,2]. The main reason for the extreme difficulty to treat the infection of bacteria is the shortage of new therapeutic methods to discover new antibacterial drugs. This situation has given rise to fears of a ‘post-antibiotic era’, as it has been approximated that 5–20 new antibacterial drugs need to enter clinical development to contend with the existing problem [3]. Hence, multiple approaches to discover antibiotics are needed.

Research has demonstrated that natural products play a dominant role in the discovery of leads for the development of drugs for the treatment of human diseases [4]. Triterpenoids, which have more than 20,000 structures identified to date are one of the most abundant classes of natural products [5,6]. The class of triterpenoids have attracted much attention due to their remarkably broad spectrum of pharmacological activities, such as anticancer, antimicrobial, antiinflammatory, immunomodulatory, antidiabetic, antioxidant, antiviral, hepato- and cardio-protective effects [7,8,9,10,11,12,13,14]. A recent article demonstrated that two new triterpenoid saponins isolated from the methanolic extract of the leaves of *Dolichandrone spathacea* possess antibacterial activity against methicillin-resistant *Staphylococcus aureus* (MRSA) [15] (Figure 1). Furthermore, the triterpenoid aglycone *α*-hederin, cationic triterpenoids **CSA** and a clinically used triterpenoid fusidic acid also have been proved to possess antibacterial activity [16] (Figure 1).

Ocotillol-type saponins are one kind of tetracyclic triterpenoids, bearing a tetrahydrofurane ring in the side chain [17,18]. Natural ocotillol-type saponins have been isolated from *Panax quinquefolius*, *Panax japonicus* L., *Hana mina* and *Vietnamese ginseng* [19]. Meng et al. found that 20(*S*)-protopanaxadiol (PPD) and 20(*R*)-ocotillol enantiopure improved the pathological changes of myocardial ischemic injury in rats [20]. Yang et al. indicated that ocotillol-type amide derivatives could overcome multidrug resistance (MDR) in cancer [21]. In our previous work, the semi-synthesis of 20(*S*/*R*)-ocotillol type saponins has been reported (Figure 1). Many of the ocotillo-type derivatives were synthesized and showed good antibacterial activity against Gram-positive bacteria by modifying the structure of the configuration. In our recent research, we designed C-3 with free aliphatic amino derivatives which exhibited good antibacterial activity against Gram-positive bacteria; compound **I** was one of these derivatives with a minimum inhibitory concentrations (MIC) value of 4 µg/mL against MRSA USA300. Besides, when combined with kanamycin and chloramphenicol, compound **I** also showed good synergistic inhibitory effects against Gram-positive bacteria [17]. Furthermore, through design and synthesis of compound **II**, we found that the ocotillol-type triterpenoid core may play a role in the bacterial membrane [22]. In addition, the preliminary structure-activity relationship was summarized thus; that hydrogen donors at C-3 positions are of great importance and -NH_2_ on the C-3 side chain is preferred for antibacterial activity [16,22,23,24]. According to our previous works, we intended to introduce free hydroxyl groups at the C-3 position, as well free amino, to verify whether this group can increase the antibacterial activity, and the antibacterial evaluations are described in the following.

## 2. Results and Discussion

### 2.1. Chemistry

Aiming at constructing a PPD skeleton, the synthetic routes were devised. We selected the acetyl group to protect C-3 and C-12 hydroxy groups, then through epoxidation by 3-chloroperoxybenzoic acid (*m*-CPBA) and removal of acetyl group reaction, the mother nuclei **3** and **4** were obtained and the specific synthetic route were shown in Scheme 1. Ocotillol-type derivatives **7**–**26** were synthesized as shown in Scheme 2. Compounds **3** and **4** were reacted with succinic anhydride to give intermediate **5** and **6**, then the isomers **5** and **6** were coupled with different side chain binding free hydroxyl in dry dichloromethane (DCM) as well as an appropriate amount of catalyst can produce ocotillol-type derivatives **7**–**14**. Similarly, target compounds **15**–**26** were obtained by binding the side chain, which amino group was protected with t-butyloxycarboryl with **5** and then deprotection using trifluoroacetic acid (TFA). The structures of all products were confirmed by ^1^H-NMR, ^13^C-NMR and HR-MS.

### 2.2. Antibacterial Activity

We tested ocotillol-type derivatives **7**–**19**, **21**–**25** for in vitro activity against five types of bacteria, including Gram-positive and Gram-negative strains, as well drug-resistant organisms (Table 1).

Among the synthesized C-3 position free hydroxyl substituted compounds **7**–**14**, the biological evaluation revealed that compounds **10** and **11** (MIC value of 128 μg/mL) had less effect against *S. aureus*. Meanwhile, target compounds **8** and **11** (MIC value of 32 μg/mL) showed moderate activity against *B. sub*, compared with lead compound **3** (MIC value of 128 μg/mL). It was noteworthy that compound **11** (MIC 32 μg/mL against *B. sub*, 32 μg/mL against *E. coli*) was found to be more potent than compound **3** against both gram-positive and gram-negative bacteria. Consequently, we concluded that the antibacterial activity of the the target compound was clearly selective, and the activity against the gram-positive bacteria (*B. sub*) were stronger than that of the gram-negative bacteria (*E. coli*, *A. baum*).

After the derivatization of different amino acids in the C-3 position, bioactivity data indicated that derivatives with 24(*R*)-configuration had significantly improved activity against Gram-positive bacteria, compared with that of constructing compound **3**. The same was true of compounds with 24(*S*)-configuration.

The data illustrated that, with unprotected primary amine groups, this series of target compounds **15**–**24** showed potent inhibitory activity against *B. sub* 168 with MIC values of about 2–16 µg/mL. Furthermore, compound **24** showed the most potent activity with MIC value of 2 µg/mL, which is equivalent to the positive drug kanamycin. The study found that the aliphatic amino acid derivatives **15**–**20** inhibited the growth of *S. aureus* in vitro with MIC values from 8 µg/mL to 32 µg/mL. Moreover, aromatic amino acid derivatives **21**–**24** had mild to good inhibitory activity against MRSA, with MIC values of about 8–32 µg/mL. in particular, compound **22** (MIC value of 2 μg/mL) had the same antibacterial activity as kanamycin.

As shown in Table 1, compounds **21** and **23**, derived from compound **3** with 24(*R*)-configuration, possessed moderate activity against two kinds of Gram-positive bacterias, whereas 24(*S*)-configuration compounds **22**, **24** showed superior antibacterial capacity. Additionally, a similar inhibitory activity against S. aureus was observed between epimers **15**, **17**, **19** and **16**, **18**, **20** and the latter is more active than the former. This showed that the C-24 position configuration had selection for activity.

The minimum inhibitory concentration demonstrated that most of the ocotillol 3-amino derivatives displayed potent antibacterial activity against gram-positive bacteria. According to previous studies [23], the speculative mechanism of cationic ocotillol-type triterpene derivatives acted mainly on bacterial cell membranes, which can change the permeability and fluidity of cell membranes or act as drug efflux pumps to increase the ability of traditional antibiotics to penetrate the cell membrane. Considering the better antibacterial activity of free amino substituted derivities, two new 3-amino derivatives were synthesised with two free aminos in the C-3 position. Initial minimum inhibitory concentration was shown in Table 1.

As shown in Table 1, the derivatives **25** and **26** displayed significantly increased antibacterial activity against gram-negative bacteria compared to the derivatives (**15**–**24**) while maintaining excellent activity against Gram-positive bacterial. It is speculated that the compounds can act on the bilayer membrane system of the gram-negative bacteria due to the increase of the free amino group. This result proved the importance of cationic amino groups for antibacterial activity once again and provide guidance for the design of novel compounds.

### 2.3. ADME Properties

Given the better antibacterial activity of two free amino substituted derivities, we further predicted absorption, distribution, metabolism, and excretion (ADME) properties.

The ADME properties were displayed in Table 2 and the results were calculated using ACD/Percepta. In order to explore the relationship between amino substitution and antibacterial activities, the ADME properties of compounds **15**–**26** were predicted. As shown in Table 2 column 2, those derivatives demonstrated similar solubility, except compounds **21**–**24** bearing aromatic groups. The logarithm of the partition coefficient (LogP) and the logarithm of the distribution constant (LogD) showed similar results. In contrast, compounds **25** and **26** showed a good solubility. Although it is a pity that the predicted value of the software can not reflect the difference in configuration. This result can also provide guidance for the design of new compounds, that is to say, the aliphatic series derivatives with multiple amino groups at C-3 position may have a better activity.

### 2.4. The Structure-Activity Relationships (SARs) of Ocotillol-Type Derivatives

Based on the previous data and our recent work, a more comprehensive structure-activity relationship could be summarized in Figure 2: (a) C-3 and C-12: hydrogen bond donors rather than hydrogen acceptors at C-3 and C-12 were important to improve activity against gram positive bacteria, once when C-3 and C-12 converted to ketones, the activity, new substituted derivatives will show decreased activity; (b) C-24: the C-24 configuration significantly affect antibacterial activity without substitution at the C-3 OH and 24(*S*)-configuration rather than 24(*R*)-configuration was preferred for compounds’ activities; and (c) C-3: substitution at C-3 OH can cause changes in molecular conformation resulting in bioactive 24(*R*)-compounds; an exception to hydrogen bond donors at C-3 can improve activities was that some nitrated derivatives at C-3 could also exhibit antibacterial activity with broader spectrum due to directed NO release on the bacterial membrane; polar aromatic substitutions at the C-3 OH also furnish activity with broader spectrum; aliphatic chain substituted derivatives displayed only mild activity against gram positive bacteria; hydroxyl was a hydrogen bond donor, however the activities of derivatives with free hydroxyl at C-3 are poor; substitution at C-3 OH with amino acid could enhanced activity against gram-positive bacteria and the antibacterial spectrum will expend when two free amino substitute at C-3 OH.

### 2.5. Pharmacophore Requirements

According to our previous studies, fourteen C-3 substituted ocotillol-type derivatives with good antibacterial activities were selected to generate pharmacophores and guide the design of novel derivatives. Galahad module of Sybyl-X 2.0 was used to generate pharmacophore using population size of 20 and maximum generations as 10. Finally, 20 models were generated. (Figure 3) The best pharmacophore model was chosen with low energy and high value of steric and hydrogen bonding. Nine pharmacophoric features, namely three acceptor atoms (AA-2,3,4) and six hydrophobic centre (HY-5,6,7,8,9,10) were identified. The two acceptor atoms were at the C-3 position, C-12 position and the furan ring. Hydrophobic centre is the fused four-membered ring itself.

## 3. Materials and Methods

### 3.1. Chemical Reagents and Instruments

Most chemicals and solvents were analytical grade and, when necessary, were purified and dried using standard methods. Melting points were measured on an XT3A micro-melting point apparatus and are uncorrected (Beijing Keyi Company, Beijing, China). ^1^H-NMR and ^13^C-NMR spectra were recorded with a Bruker AV-400 instrument or a Bruker AV-300 (Bruker, Ettlingen, Germany) in the indicated solvents (Tetramethyl silane TMS as internal standard), the values of the chemical shifts expressed in *δ* values (ppm) and the coupling constants (*J*) in Hz. High resolution mass spectra were measured using an Agilent QTOF 6520 (Agilent, Palo Alto, CA, USA) and Q-Exactive Orbtitrap MS system (Thermo Scientific, Waltham, MA, USA).

### 3.2. General Procedure for the Synthesis of Compounds ***3**–**6***

The synthesis of compounds **3**, **4** used PPD as the starting material with three steps of acid anhydride protection, epoxidation and base deprotection. The organic layer was washed with water and brine and dried over anhydrous sodium sulfate, purified over silica gel (Scheme 1). In the stirred solution of **3** and **4**, butanedioic anhydride and DMAP were added at 35 °C for 5 h to afford compounds **5** and **6**. The specific synthesis method of compounds **3**–**6** were according to the published procedures [16,22].

### 3.3 General Procedure for the Synthesis of Compounds ***7**–**14***

Diols (0.071 mmol) was added to a solution of **5** or **6** (0.035 mmol) in dry dichloromethane (10 mL), then DMAP (0.05 mmol) and 1-ethyl-3-(3-dimethylaminopropyl) carbodiimide (EDCI) (0.05 mmol) were added. After stirring at room temperature for 4 h, the reaction mixture was washed with 10% hydrochloric acid, water and saturated brine, dried over anhydrous sodium sulfate and concentrated. The organic mixture was purified by column chromatography over silica gel (petroleum ether: acetone = 12:1–8:1). ^1^H-NMR, ^13^C-NMR and HR-MS spectra of compounds are shown in Appendix A.

*4-[(20S,24R)-Epoxy-12β,25-diol-dammarane-3β-O]-4-oxo-butyryloxyethanol* (**7**). White solid (yield 83.8%); m.p. 39–40 °C; HR-MS (HESI) *m/z* calcd. for C_36_H_60_O_8_ [M + H]^+^: 621.43610, found: 621.43286; ^1^H-NMR (CDCl_3_ 400 MHz) *δ* (ppm): 0.83 (6H, s, -CH_3_), 0.84 (6H, s, -CH_3_), 0.87 (3H, s, -CH_3_), 0.89 (3H, s, -CH_3_), 0.97 (3H, s, -CH_3_), 1.09 (3H, s, -CH_3_), 1.25 (3H, s, -CH_3_), 1.27 (3H, s, -CH_3_), 2.66 (4H, s, -CH_2_CH_2_-), 3.47 (1H, td, *J* = 10.4 Hz, *J* = 4.5 Hz), 3.80 (1H, td, *J* = 8.6 Hz, *J* = 6.8 Hz; 2H, t, *J* = 4.5 Hz), 4.23 (2H, t, *J* = 4.5 Hz), 4.47 (1H, dd, *J* = 8.6 Hz, *J* = 7.7 Hz); ^13^C-NMR (CDCl_3_, 100 MHz) *δ* (ppm): 26.09, 27.58, 27.90, 28.56, 29.30, 29.34, 29.63, 29.67, 31.17, 31.28, 31.90, 32.58, 34.69, 37.02, 37.88, 38.56, 39.72, 47.91, 49.34, 50.36, 51.97, 55.99, 61.04, 66.30, 70.09, 70.91, 81.40, 85.38, 86.49, 172.23, 172.53.

*4-[(20S,24S)-Epoxy-12β,25-diol-dammarane-3β-O]-4-oxo-butyryloxyethanol* (**8**). White solid (yield 84.9%); m.p. 73–76 °C; HR-MS (HESI) *m/z* calcd. for C_36_H_60_O_8_ [M + H]^+^: 621.43610, found: 621.43457; ^1^H-NMR (CDCl_3_, 400 MHz) *δ* (ppm): 0.84 (3H, s, -CH_3_), 0.85 (6H, s, -CH_3_), 0.90 (3H, s, -CH_3_), 1.01 (3H, s, -CH_3_), 1.10 (3H, s, -CH_3_), 1.22 (3H, s, -CH_3_), 1.27 (3H, s, -CH_3_), 2.66 (4H, s, -CH_2_CH_2_-), 3.49 (1H, td, *J* = 10.3 Hz, *J* = 4.7 Hz), 3.79 (2H, t, *J* = 4.7 Hz), 3.85 (1H, dd, *J* = 10.7 Hz, *J* = 5.3 Hz), 4.22 (2H, t, *J* = 4.5 Hz), 4.48 (1H, dd, *J* = 9.2 Hz, *J* = 7.2 Hz), 5.7 (1H, s); ^13^C-NMR (CDCl_3_, 100 MHz) *δ* (ppm): 24.15, 25.01, 27.88, 27.91, 28.47, 28.81, 29.25, 29.58, 31.56, 31.58, 32.14, 34.60, 37.00, 37.85, 38.49, 39.68, 48.72, 48.83, 50.05, 52.09, 55.92, 60.90, 66.26, 70.02, 70.42, 81.35, 87.09, 87.32, 172.17, 172.5.

*4-[(20S,24R)-Epoxy-12β,25-diol-dammarane-3β-O]-4-oxo-butyryloxybutanol* (**9**). White solid (yield 84.8%); m.p. 54–57 °C; HR-MS (HESI) *m/z* calcd. for C_38_H_64_O_8_ [M + H]^+^: 649.46740, found: 649.46625; ^1^H-NMR (CDCl_3_, 400 MHz) *δ* (ppm): 0.84 (6H, s, -CH_3_), 0.87 (3H, s, -CH_3_), 0.89 (3H, s, -CH_3_), 0.98 (3H, s, -CH_3_), 1.09 (3H, s, -CH_3_), 1.26 (3H, s, -CH_3_), 1.27 (3H, s, -CH_3_), 2.62 (4H, s, -CH_2_CH_2_-), 3.47 (1H, td, *J* = 10.2 Hz, *J* = 4.4 Hz), 3.66 (2H, t, *J* = 6.4 Hz), 3.82 (1H, td, *J* = 8.6 Hz, *J* = 6.8 Hz), 4.11 (2H, t, *J* = 6.4 Hz), 4.46 (1H, dd, *J* = 10.3 Hz, *J* = 6.2 Hz); ^13^C-NMR (CDCl_3_, 100 MHz) *δ* (ppm): 26.13, 27.58, 27.90, 27.93, 28.58, 29.09, 29.26, 29.56, 29.68, 31.19, 31.33, 32.60, 34.74, 37.06, 37.90, 38.60, 39.75, 47.95, 49.39, 50.40, 52.00, 56.06, 62.33, 64.50, 70.09, 70.93, 81.21, 85.41, 86.51, 171.98, 172.36.

*4-[(20S,24S)-Epoxy-12β,25-diol-dammarane-3β-O]-4-oxo-butyryloxybutanol* (**10**). White solid (yield 81%); m.p. 57–59 °C; HR-MS (HESI) *m/z* calcd. for C_38_H_64_O_8_ [M + H]^+^: 649.46740, found: 649.46637; ^1^H-NMR (CDCl_3_, 400 MHz) *δ* (ppm): 0.85 (6H, s, -CH_3_), 0. 90 (3H, s, -CH_3_), 0.98 (3H, s, -CH_3_), 1.01 (3H, s, -CH_3_), 1.10 (3H, s, -CH_3_), 1.22 (3H, s, -CH_3_), 1.25 (3H, s, -CH_3_), 1.27 (3H, s, -CH_3_), 2.62 (4H, s, -CH_2_CH_2_-), 3.49 (1H, td, *J* = 10.2 Hz, *J* = 4.6 Hz), 3.64 (2H, t, *J* = 6.4 Hz) 3.85 (1H, td, *J* = 10.7 Hz, *J* = 5.3 Hz), 4.11 (2H, t, *J* = 6.3 Hz), 4.47 (1H, dd, *J* = 10.3 Hz, *J* = 6.0 Hz); ^13^C-NMR (CDCl_3_, 100 MHz) *δ* (ppm): 28.46, 28.80, 29.01, 29.19, 29.29, 29.49, 29.62, 29.95, 31.54, 31.58, 31.85, 32.13, 32.69, 34.60, 36.99, 37.36, 37.85, 38.48, 39.67, 48.71, 48.82, 50.04, 52.08, 55.93, 62.13, 64.47, 69.99, 70.40, 81.14, 87.07, 87.31, 171.92, 172.32, 171.32, 172.32.

*4-[(20S,24R)-Epoxy-12β,25-diol-dammarane-3β-O]-4-oxo-butyryloxybutynol* (**11**). White solid (yield 85.6%); m.p. 40 °C; HR-MS (HESI) *m/z* calcd. for C_38_H_60_O_8_ [M + H]^+^: 645.43610, found: 645.43481; ^1^H-NMR (CDCl_3_, 400 MHz) *δ* (ppm): 0.84 (3H, s, -CH_3_), 0.87 (3H, s, -CH_3_), 0.89 (3H, s, -CH_3_), 0.97 (3H, s, -CH_3_), 1.10 (3H, s, -CH_3_), 1.25 (3H, s, -CH_3_), 1.26 (3H, s, -CH_3_), 1.27 (3H, s, -CH_3_), 2.62 (4H, m, -CH_2_CH_2_-), 3.47 (1H, td, *J* = 10.4 Hz, *J* = 4.5 Hz), 3.82 (1H, td), 4.29 (2H, s), 4.46 (1H, dd, *J* = 10.1 Hz, *J* = 6.5 Hz), 4.73 (2H, s); ^13^C-NMR (CDCl_3_, 100 MHz) *δ* (ppm): 26.10, 27.57, 27.88, 27.92, 28.56, 29.04, 29.38, 29.67, 31.18, 31.31, 31.90, 32.59, 34.72, 37.05, 37.88, 38.58, 39.74, 47.93, 49.36, 50.38, 50.97, 51.99, 52.45, 56.04, 70.11, 70.92, 79.59, 81.33, 85.13, 85.39, 86.50, 171.62, 171.74.

*4-[(20S,24S)-Epoxy-12β,25-diol-dammarane-3β-O]-4-oxo-butyryloxybutynol* (**12**). White solid (yield 80.7%); m.p. 64 °C; HR-MS (HESI) *m/z* calcd. for C_38_H_60_O_8_ [M + H]^+^: 645.43610, found: 645.43494; ^1^H-NMR (CDCl_3_, 400 MHz) *δ* (ppm): 0.85 (6H, s, -CH_3_), 0.90 (3H, s, -CH_3_), 1.01 (3H, s, -CH_3_), 1.10 (3H, s, -CH_3_), 1.23 (3H, s, -CH_3_), 1.25 (3H, s, -CH_3_), 1.27 (3H, s, -CH_3_), 2.62 (4H, m, -CH_2_CH_2_-), 3.49 (1H, td, *J* = 10.3 Hz, *J* = 4.7 Hz), 3.82 (1H, td, *J* = 10.8 Hz, *J* = 5.3 Hz), 4.28 (2H, t, *J* = 1.7 Hz), 4.48 (1H, dd, *J* = 10.3 Hz, *J* = 6.1 Hz), 4.73 (2H, t, *J* = 3.4 Hz), 5.8 (2H, s); ^13^C-NMR (CDCl_3_, 100 MHz) *δ* (ppm): 27.91, 27.97, 28.50, 28.84, 29.02, 29.36, 29.66, 31.58, 31.62, 31.89, 32.18, 34.63, 37.04, 37.87, 38.51, 39.71, 48.75, 48.85, 50.08, 50.92, 52.12, 52.46, 55.97, 70.04, 70.45, 79.51, 81.32, 85.18, 87.13, 87.35, 128.80, 171.64, 171.73.

*4-[(20S,24R)-Epoxy-12β,25-diol-dammarane-3β-O]-4-oxo-butyryloxy phenol* (**13**). White solid (yield 83%); m.p. 68 °C; HR-MS (HESI) *m/z* calcd. for C_40_H_60_O_8_ [M + H]^+^: 669.43610, found: 669.43536; ^1^H-NMR (CDCl_3_, 400 MHz) *δ* (ppm): 0.84 (3H, s, -CH_3_), 0.85 (3H, s, -CH_3_), 0.89 (3H, s, -CH_3_), 0.97 (3H, s, -CH_3_), 1.10 (3H, s, -CH_3_), 1.25 (3H, s, -CH_3_), 1.26 (3H, s, -CH_3_), 1.28 (3H, s, -CH_3_), 2.71 (2H, t, -CH_2_-), 2.83, (2H, t, -CH_2_-), 3.51 (1H, td, *J* = 8.4 Hz), 3.82 (1H, td, *J* = 6.8 Hz), 4.49 (1H, dd, *J* = 10.1 Hz, *J* =5.0 Hz), 6.78 (2H, Ph), 6.90 (2H, Ph); ^13^C-NMR (CDCl_3_, 100 MHz) *δ* (ppm): 26.06, 27.62, 27.90, 27.98, 28.58, 29.37, 29.52, 29.68, 30.02, 31.21, 31.31, 31.91, 32.62, 34.73, 37.04, 37.92, 38.58, 39.75, 47.95, 49.38, 50.41, 52.02, 56.05, 70.32, 71.01, 81.40, 85.38, 86.54, 115.92, 122.32, 143.95, 153.63, 171.32, 171.85.

*4-[(20S,24S)-Epoxy-12β,25-diol-dammarane-3β-O]-4-oxo-butyryloxy phenol* (**14**). White solid (yield 79%); m.p. 88 °C; HR-MS (HESI) *m/z* calcd. for C_40_H_60_O_8_ [M + H]^+^: 669.43610, found: 669.43536; ^1^H-NMR (CDCl_3_, 400 MHz) *δ* (ppm): 0.84 (3H, s, -CH_3_), 0.86 (3H, s, -CH_3_), 0.89 (3H, s, -CH_3_), 0.91 (3H, s, -CH_3_), 1.01 (3H, s, -CH_3_), 1.10 (3H, s, -CH_3_), 1.25 (3H, s, -CH_3_), 1.27 (3H, s, -CH_3_), 2.71 (2H, t, -CH_2_-), 2.84 (2H, t, -CH_2_-), 3.52 (1H, td, *J* = 8.4 Hz), 3.85 (1H, dd, *J* = 10.8 Hz, *J* = 5.3 Hz), 4.50 (1H, dd, *J* = 5.8 Hz, *J* = 10.6 Hz), 5.95 (H, s, OH), 6.17 (H, s, OH), 6.78 (2H, Ph), 6.90 (2H, Ph); ^13^C-NMR (CDCl_3_, 100 MHz) *δ* (ppm): 24.11, 25.02, 27.95, 27.97, 28.50, 28.82, 28.31, 29.46, 29.67, 31.54, 31.57, 32.19, 34.61, 37.01, 37.88, 38.48, 39.69, 48.68, 48.80, 50.06, 52.13, 55.94, 70.10, 70.58, 81.37, 87.18, 87.36, 115.92, 122.28, 143.78, 153.74, 171.41, 171.89.

### 3.4. General Procedure for the Synthesis of Compounds ***15**–**26***

To a solution of **3** or **4** in dry dichloromethane (10 mL), Boc-amino acids (0.21 mmol), DMAP (0.05 mmol) and EDCI (0.15 mmol) were added, and the mixture was stirred at room temperature for 7 h. The reaction mixture was washed with water and saturated brine, dried over anhydrous sodium sulfate, filtered, and concentrated. The organic mixture was purified by column chromatography over silica gel (trichloromethane:methanol = 250:1). The above intermediate (0.036 mmol) was dissolved in ethyl acetate-35% HCl (10 mL, 20:1), then stirred at 35 °C for 2 h. The reaction solution was washed with saturated sodium bicarbonate, water, and saturated brine in that order, dried over anhydrous sodium sulfate, filtered, and concentrated. The organic mixture was purified by column chromatography over silica gel (trichloromethane: methanol = 200:1) to provide compounds **15**–**20**.

The intermediate of Boc-amino acids at C-3 was directly dissolved in dry dichloromethane (8 mL), trifluoroacetic acid (1 mL) was added, and the mixture was stirred at room temperature for 2 h. The reaction solution was washed with saturated sodium bicarbonate, water, and saturated brine in that order, dried over anhydrous sodium sulfate, filtered, and concentrated. Silica gel column chromatography (dichloromethane:methanol = 80:1–10:1) gave the desired product **21**–**26**.

*(20S,24R)-Epoxy-3β-O-(2-aminopropionyl)-dammarane-12β,25-diol* (**15**). Light yellow solid (90.2%); m.p. 78–80 °C; HR-MS (HESI) *m/z* calcd. for C_33_H_57_NO_5_ [M + H]^+^: 548.4315, found: 548.4321; ^1^H-NMR (CDCl_3_, 300 MHz) *δ* (ppm): 0.77(3H, s, -CH_3_), 0.79 (3H, s, -CH_3_), 0.81 (3H, s, -CH_3_), 0.83 (3H, s, -CH_3_), 0.91 (3H, s, -CH_3_), 1.02 (3H, s, -CH_3_), 1.19 (3H, s, -CH_3_), 1.20 (3H, s, -CH_3_), 1.37 (3H, s, -CH_3_), 3.43 (1H, dt, *J* = 10.2 Hz, *J* = 4.7 Hz, -OCH-), 3.58 (1H, m, -NCH-), 3.78 (1H, dd, *J* = 10.8 Hz, *J* = 5.4 Hz, -OCH-), 4.45 (1H, dd, *J* = 10.4 Hz, *J* = 6.0 Hz, -OCH-); ^13^C-NMR (CDCl_3_, 75 MHz) *δ* (ppm): 15.38, 16.36, 16.48, 18.13, 18.17, 20.08, 23.68, 25.00, 26.13, 27.60, 27.91, 28.00, 28.59, 29.70, 31.19, 31.33, 32.61, 34.72, 37.06, 38.05, 38.56, 39.74, 47.94, 49.37, 50.39, 52.00, 56.01, 70.11, 70.92, 81.73, 85.41, 86.51, 175.21.

*(20S,24S)-Epoxy-3β-O-(2-aminopropionyl)-dammarane-12β,25-diol* (**16**). Light yellow solid (82.1%); m.p. 79–80 °C; HR-MS (HESI) *m/z* calcd. for C_33_H_57_NO_5_ [M + H]^+^: 548.4315, found: 548.4326; ^1^H-NMR (CDCl_3_, 300 MHz) *δ* (ppm): 0.85 (3H, s, -CH_3_), 0.87 (3H, s, -CH_3_), 0.92 (6H, s, -CH_3_), 1.02 (3H, s, -CH_3_), 1.11 (3H, s, -CH_3_), 1.23 (3H, s, -CH_3_), 1.28 (3H, s, -CH_3_), 1.40 (3H, s, -CH_3_), 3.53 (1H, dt, *J* = 10.2 Hz, *J* = 4.7 Hz, -OCH-), 3.58 (1H, m, -NCH-), 3.88 (1H, dd, *J* = 10.8 Hz, *J* = 5.4 Hz, -OCH-), 4.53 (1H, dd, *J* = 10.4 Hz, *J* = 6.0 Hz, -OCH-), 5.80 (1H, s, -OH); ^13^C-NMR (CDCl_3_, 75 MHz) *δ* (ppm): 15.45, 16.32, 16.50, 17.77, 18.14, 20.29, 23.70, 24.18, 25.08, 26.13, 27.60, 27.91, 28.53, 28.87, 31.60, 31.68, 32.20, 34.65, 37.07, 38.05, 38.52, 39.74, 48.78, 48.88, 50.11, 52.14, 55.97, 70.08, 70.46, 81.54, 87.13, 87.39, 175.23.

*(20S,24R)-Epoxy-3β-O-[(2-amino-4-methyl)pentanoyl)-dammarane-12β,25-diol* (**17**). Light yellow solid (76.2%); m.p. 79–81 °C; HR-MS (ESI) *m/z* calcd. for C_36_H_63_NO_5_ [M + H]^+^: 590.4784, found: 590.4779; ^1^H-NMR (CDCl_3_, 300 MHz) *δ* (ppm): 0.75 (3H, s, -CH_3_), 0.77 (3H, s, -CH_3_), 0.79 (3H, s, -CH_3_), 0.80 (3H, s, -CH_3_), 0.85 (3H, s, -CH_3_), 0.89 (3H, s, -CH_3_), 1.00 (3H, s, -CH_3_), 1.17 (3H, s, -CH_3_), 1.18 (3H, s, -CH_3_), 1.37 (3H, s, -CH_3_), 3.43 (1H, dt, *J* = 10.2 Hz, *J* = 4.7 Hz, -OCH-), 3.45 (1H, m, -NCH-), 3.74 (1H, dd, *J* = 10.8 Hz, *J* = 5.4 Hz, -OCH-), 4.45 (1H, dd, *J* = 10.4 Hz, *J* = 6.0 Hz, -OCH-); ^13^C-NMR (CDCl_3_, 75 MHz) *δ* (ppm): 15.38, 16.38, 16.55, 18.17, 20.08, 22.15, 23.67, 24.81, 25.00, 26.14, 27.60, 27.92, 28.09, 28.59, 29.70, 31.19, 31.33, 32.60, 34.74, 37.07, 37.96, 38.05, 38.53, 39.75, 43.75, 47.94, 49.36, 50.40, 52.01, 56.07, 70.11, 70.94, 81.73, 85.41, 86.51, 175.20.

*(20S,24S)-Epoxy-3β-O-[(2-amino-4-methyl)pentanoyl)-dammarane-12β,25-diol* (**18**). Light yellow solid (77.1%); m.p. 76–78 °C; HR-MS (ESI) *m/z* calcd. for C_36_H_63_NO_5_ [M + H]^+^: 590.4784, found: 590.4778; ^1^H-NMR (CDCl_3_, 300 MHz) *δ* (ppm): 0.79 (3H, s, -CH_3_), 0.80 (3H, s, -CH_3_), 0.85 (6H, s, -CH_3_), 0.88 (6H, s, -CH_3_), 0.94 (3H, s, -CH_3_), 1.04 (3H, s, -CH_3_), 1.16 (3H, s, -CH_3_), 1.21 (3H, s, -CH_3_), 3.43 (1H, dt, *J* = 10.2 Hz, *J* = 4.7 Hz, -OCH-), 3.45 (1H, m, -NCH-), 3.81 (1H, dd, *J* = 10.8 Hz, *J* = 5.4 Hz, -OCH-), 4.45 (1H, dd, *J* = 10.4 Hz, *J* = 6.0 Hz, -OCH-); ^13^C-NMR (CDCl_3_, 100 MHz) *δ* (ppm): 15.38, 16.38, 16.55, 18.17, 20.08, 22.15, 23.67, 24.81, 25.00, 26.14, 27.60, 27.92, 28.09, 28.59, 29.70, 31.19, 31.33, 32.60, 34.74, 37.07, 37.96, 38.05, 38.53, 39.75, 43.75, 47.94, 49.36, 50.40, 52.01, 56.07, 70.11, 70.94, 81.73, 87.10, 87.33, 175.23.

*(20S,24R)-Epoxy-3β-O-[(2-amino-3-methyl)butanoyl)-dammarane-12β,25-diol* (**19**). Light yellow solid (79.9%); m.p. 77–79 °C; HR-MS (ESI) *m/z* calcd. for C_36_H_63_NO_5_ [M + H]^+^: 576.4628, found: 576.4627; ^1^H-NMR (CDCl_3_, 400 MHz) *δ* (ppm): 0.87(6H, s, -CH_3_), 0.89 (6H, s, -CH_3_), 0.91 (6H, s, -CH_3_), 1.01 (3H, s, -CH_3_), 1.10 (3H, s, -CH_3_), 1.23 (3H, s, -CH_3_), 1.28 (3H, s, -CH_3_), 3.28 (1H, d, *J* = 4.5, -OCH-), 3.53 (1H, dt, *J* = 10.2 Hz, *J* = 4.7 Hz, -NCH-), 3.88 (1H, dd, *J* = 10.8 Hz, *J* = 5.4 Hz, -OCH-), 4.52 (1H, dd, *J* = 10.4 Hz, *J* = 6.0 Hz, -OCH-); ^13^C-NMR (CDCl_3_, 100 MHz) *δ* (ppm): 15.33, 16.38, 16.43, 18.12, 19.22, 20.22, 23.18, 23.63, 24.89, 26.08, 27.56, 27.87, 28.30, 28.53, 29.65, 31.14, 31.27, 32.57, 34.66, 37.00, 37.99, 38.51, 39.69, 47.88, 49.32, 50.34, 51.95, 52.64, 55.96, 70.08, 70.88, 81.53, 85.35, 86.46, 175.52.

*(20S,24S)-Epoxy-3β-O-[(2-amino-3-methyl)butanoyl)-dammarane-12β,25-diol* (**20**). Light yellow solid (82.2%); m.p. 79–81 °C; HR-MS (ESI) *m/z* calcd. for C_36_H_63_NO_5_ [M + H]^+^: 576.4628, found: 576.4636; 576.4636; ^1^H-NMR (CDCl_3_, 400 MHz) *δ* (ppm): 0.87(6H, s, -CH_3_), 0.90 (6H, s, -CH_3_), 1.01 (6H, s, -CH_3_), 1.10 (6H, s, -CH_3_), 1.23 (3H, s, -CH_3_), 1.28 (3H, s, -CH_3_), 3.28 (1H, d, *J*=4.5, -OCH-), 3.53 (1H, dt, *J* = 10.2 Hz, *J* = 4.7 Hz, -NCH-), 3.88 (1H, dd, *J* = 10.8, *J* = 5.4 Hz, -OCH-), 4.52 (1H, dd, *J* = 10.4 Hz, *J* = 6.0 Hz, -OCH-); ^13^C-NMR (CDCl_3_, 100 MHz) *δ* (ppm): 15.42, 16.28, 16.59, 16.69, 17.72, 18.14, 19.64, 23.75, 24.21, 25.00, 28.00, 28.03, 28.50, 28.84, 31.56, 31.64, 31.74, 32.16, 34.64, 37.02, 37.88, 38.50, 39.70, 48.76, 48.87, 50.08, 52.11, 55.98, 60.31, 69.98, 70.82, 81.28, 87.10, 87.33, 175.20.

*(20S,24R)-Epoxy-3β-O-(2S-amino-phenylpropionyl)-dammarane-12β,25-diol* (**21**). White solid (yield 45%); m.p. 230–233 °C; HR-MS (HESI) *m/z* calcd. for C_39_H_61_NO_5_ [M + H]^+^: 624.46225, found: 624.46115; ^1^H-NMR (CDCl_3_, 300 MHz) *δ* (ppm) 0.76 (3H, s, -CH_3_), 0.78 (3H, s, -CH_3_), 0.85 (3H, s, -CH_3_), 0.93 (3H, s, -CH_3_), 1.05 (3H, s, -CH_3_), 1.16 (3H, s, -CH_3_), 1.18 (3H, s, -CH_3_), 1.22 (3H, s, -CH_3_), 2.21–2.14 (1H, m, -CH_2_-), 2.74 (1H, dd, *J* = 13.3 Hz, 8.5 Hz, -CH_2_-), 3.10 (1H, dd, *J* = 13.5 Hz, 4.9 Hz, -CH_2_-), 3.46 (1H, td, *J* = 10.3 Hz, 4.6 Hz, -OCH-), 3.66 (1H, t, *J* = 10.4 Hz, -NCH-), 3.83 (1H, dd, *J* = 11.2 Hz, 5.4 Hz, -OCH-), 4.47 (1H, dd, *J* = 10.3 Hz, 5.9 Hz, -OCH-), 7.23–7.15 (5H, m, Ar-H). ^3^C-NMR (CDCl_3_, 100 MHz) *δ* (ppm) 13.81. 15.47, 16.44, 16.57, 18.23, 19.26, 23.72, 25.08, 26.21, 27.67, 27.99, 28.07, 28.67, 30.65, 31.28, 31.42, 32.69, 34.82, 37.14, 38.04, 38.68, 39.85, 48.04, 49.47, 50.50, 52.09, 56.14, 65.65, 71.02, 70.21, 81.93, 85.49, 86.59, 126.93, 128.69, 129.40, 137.25, 167.80.

*(20S,24S)-Epoxy-3β-O-(2S-amino-phenylpropionyl)-dammarane-12β,25-diol* (**22**). White solid (yield 43%); m.p. 236–240 °C; HR-MS (HESI) *m/z* calcd. for C_39_H_61_NO_5_ [M + H]^+^: 624.46225, found: 624.46149; ^1^H-NMR (CDCl_3_, 300 MHz) *δ* (ppm) 0.71 (3H, s, -CH_3_), 0.78 (3H, s, -CH_3_), 0.84 (3H, s, -CH_3_), 0.94 (3H, s, -CH_3_), 1.03 (3H, s, -CH_3_), 1.17 (3H, s, -CH_3_), 1.19 (3H, s, -CH_3_), 1.21 (3H, s, -CH_3_), 2.21–2.14 (1H, m, -CH_2_-), 2.73 (1H, dd, *J* = 13.4 Hz, 8.4 Hz, -CH_2_-), 3.09 (1H, dd, *J* = 13.4 Hz, 4.9 Hz, -CH_2_-), 3.46 (1H, td, *J* = 10.4 Hz, 4.7 Hz, -OCH-), 3.65 (1H, t, *J* = 10.0 Hz, -NCH-), 3.81 (1H, dd, *J* = 11.0 Hz, 5.6 Hz, -OCH-), 4.46 (1H, dd, *J* = 10.4 Hz, 5.8 Hz, -OCH-), 7.26–7.14 (5H, m, Ar-H).^13^C-NMR (CDCl_3_, 100 MHz) *δ* (ppm) 13.81. 15.55, 16.40, 16.59, 17.84, 19.26, 23.74, 24.27, 25.18, 28.08, 28.96, 30.65, 31.71, 32.29, 34.76, 37.16, 38.04, 38.64, 39.84, 48.89, 49.00, 50.22, 52.23, 56.10, 65.65, 70.18, 70.56, 81.96, 87.22, 87.48, 126,94, 128.70, 129.40, 137.21, 167.79.

*(20S,24R)-Epoxy-3β-O-(2S-amino-3-mercaptopropanoyl)-dammarane-12β,25-diol* (**23**). White solid (yield 39%); m.p. 235–237 °C; HR-MS (HESI) *m/z* calcd. for C_41_H_62_N_2_O_5_ [M + H]^+^: 663.47370, found: 663.47272; ^1^H-NMR (CDCl_3_, 300 MHz) *δ* (ppm) 0.78 (3H, s, -CH_3_); 0.84 (3H, s, -CH_3_), 0.87 (3H, s, -CH_3_), 0.90 (3H, s, -CH_3_), 0.98 (3H, s, -CH_3_), 1.09 (3H, s, -CH_3_), 1.27 (3H, s, -CH_3_), 1.28 (3H, s, -CH_3_), 2.19 (1H, t, *J* = 8.8 Hz, -CH_2_-), 2.95 (1H, dd, *J* = 14.3 Hz, 8.6 Hz, -CH_2_-), 3.35 (1H, dd, *J* = 14.3 Hz, 4.4 Hz, -CH_2_-), 3.52 (1H, dd, *J* = 10.4 Hz, 6.1 Hz, -OCH-), 3.85 (2H, t, *J* = 8.6 Hz, -OCH-), 4.52 (1H, t, *J* = 8.7 Hz, -OCH-), 5.61 (1H, s, Ar-H), 7.16 (2H, td, *J* = 14.9 Hz, 7.0 Hz, Ar-H), 7.36 (1H, d, *J* = 7.9 Hz, -NH-), 7.64 (1H, d, *J* = 7.6 Hz, Ar-H).^13^C-NMR (CDCl_3_, 100 MHz) *δ* (ppm) 15.47, 16.44, 16.59, 17.83, 18.24, 23.63, 24.25, 25.09, 26.18, 27.66, 27.99, 28.44, 29.78, 31.27, 32.69, 34.81, 37.13, 38.08, 39.84, 48.03, 49.45, 50.49, 52.10, 52.23, 56.13, 70.19, 70.26, 71.03, 82.42, 85.49, 86.60, 111.46, 118.67, 119.70, 122.36, 123.61, 127.30, 128.44, 129.45, 129.63, 136.21, 136.45, 170.97.

*(20S,24S)-Epoxy-3β-O-(2S-amino-3-mercaptopropanoyl)-dammarane-12β,25-diol* (**24**). White solid (yield 41%); m.p. 225–229 °C; HR-MS (HESI) *m/z* calcd. for C_41_H_62_N_2_O_5_ [M + H]^+^: 663.47370, found: 663.47284; ^1^H-NMR (CDCl_3_, 300 MHz) *δ* (ppm) 0.79 (3H, s, -CH_3_), 0.85 (3H, s, -CH_3_), 0.91 (3H, s, -CH_3_), 1.01 (3H, s, -CH_3_), 1.10 (3H, s, -CH_3_), 1.23 (3H, s, -CH_3_), 1.25 (3H, s, -CH_3_), 1.28 (3H, s, -CH_3_), 2.25 (1H, m, -CH_2_-), 2.95 (1H, dd, *J* = 14.2 Hz, 8.6 Hz, -CH_2_-), 3.35 (1H, dd, *J* = 14.3 Hz, 4.5 Hz, -CH_2_-), 3.53 (1H, td, *J* = 10.2 Hz, 5.6 Hz, -OCH-), 3.86 (2H, m, -OCH-), 4.53 (1H, t, *J* = 9.6 Hz, -OCH-), 5.76 (1H, s, Ar-H), 7.15 (2H, m, Ar-H), 7.36 (1H, d, *J* = 7.8 Hz, Ar-H), 7.64 (1H, d, *J* = 7.7Hz, -NH-).^13^C-NMR (CDCl_3_, 100 MHz) *δ* (ppm) 15.54, 16.42, 16.64, 17.86, 18.24, 23.76, 24.31, 25.14, 28.06, 28.14, 28.64, 28.97, 29.42, 29.80, 31.19, 31.70, 32.29, 34.75, 37.15, 38.06, 38.63, 39.83, 48.88, 48.98, 50.21, 52.24, 55.35, 56.09, 70.14, 70.56, 81.60, 87.24, 87.46, 111.33, 111.53, 118.88, 119.59, 122.26, 123.02, 127.49, 136.40, 174.92.

*(20S,24R)-Epoxy-3β-O-(L-lysyl)-dammarane-12β,25-diol* (**25**). White solid (yield 52%); m.p. 201–203 °C, HR-MS (HESI) *m/z* calcd. for C_36_H_64_N_2_O_5_ [M + H]^+^: 605.48880, found: 605.48920; ^1^H-NMR (CDCl_3_, 400 MHz) *δ* (ppm) 0.82 (3H, s, -CH_3_), 0.84 (3H, s, -CH_3_), 0.87 (3H, s, -CH_3_), 0.89 (3H, s, -CH_3_), 0.97 (3H, s, -CH_3_), 1.08 (3H, s, -CH_3_), 1.25 (3H, s, -CH_3_), 1.26 (3H, s, -CH_3_), 2.25 (1H, m, NCH_2_-), 2.95 (1H, dd, *J* = 14.2 Hz, 8.6 Hz, NCH_2_-), 3.36 (2H, dd, *J* = 12.0 Hz, 4.0 Hz, -NCH_2_-), 3.49 (2H, dt, *J* = 10.3 Hz, 4.5 Hz, -OCH-), 3.83 (1H, m, -OCH-), 4.49 (1H, t, *J* = 8.3 Hz, -OCH-).^13^C-NMR (CDCl_3_, 100 MHz) *δ* (ppm) 15.46. 16.43, 16.65, 18.22, 22.85, 23.80, 25.06, 26.20, 27.65, 27.96, 28.14, 28.65, 31.26, 31.41, 32.67, 34.29, 34.81, 37.13, 38.06, 38.66, 39.83, 41.17, 49.02, 49.45, 50.47, 52.08, 54.56, 56.10, 70.18, 70.99, 81.52, 85.47, 86.57, 175.68.

*(20S,24S)-Epoxy-3β-O-(L-lysyl)-dammarane-12,25β-diol* (**26**). White solid (yield 47%); m.p. 191–194 °C, HR-MS (HESI) *m/z* calcd. for C_36_H_64_N_2_O_5_ [M + H]^+^: 605.48880, found: 605.48895; ^1^H-NMR (CDCl_3_, 400 MHz) *δ* (ppm) 0.82 (3H, s, -CH_3_), 0.84 (3H, s, -CH_3_), 0.88 (6H, s, -CH_3_), 0.88 (6H, s, -CH_3_), 0.98 (3H, s, -CH_3_), 1.07 (3H, s, -CH_3_), 1.20 (6H, s, -CH_3_), 1.22 (3H, s, -CH_3_), 1.24 (3H, s, -CH_3_), 2.67 (2H, t, *J* = 6.6 Hz, NCH_2_-), 3.39 (1H, dd, *J* = 7.3 Hz, *J* = 5.2 Hz, -NCH-), 3.50 (1H, dt, *J* = 10.2 Hz, *J* = 5.1 Hz -OCH-), 3.84 (1H, dd, *J* = 10.8 Hz, *J* = 5.4 Hz -OCH-), 4.49 (1H, t, *J* = 12.0 Hz, -OCH-)^13^C-NMR (CDCl_3_, 100 MHz) *δ* (ppm) 15.56, 16.41, 16.68, 17.91, 22.10, 23.69, 23.80, 24.18, 26.91, 27.97, 28.16, 28.96, 29.78, 31.75, 31.94, 32.00, 32.33, 34.77, 34.84, 37.14, 38.05, 38.59, 39.41, 39.84, 48.87, 49.00, 50.19, 52.24, 53.61, 56.01, 70.26, 70.56, 82.75, 87.20, 87.53, 172.95.

### 3.5. Pharmacology

Luria-Bertani (LB) medium (100 μL per well) was added to the sterilized 96-well plate. B2, C2, and D2 wells were used as healthy bacterial controls, and E11, F11, and G11 wells were used as blank controls. 0.4 μL DMSO was added to the B3, C3, and D3 wells as a solvent control. The compounds were dissolved to 50 mM in DMSO and 0.4 μL of these compound solutions was added to the remaining wells (B4-G10). Then, 5 μL of the bacterial solution freshly grown in LB culture medium with an absorbance of 0.6–1.0 on the same day was added to each well of B2-G10, incubated for 18 h at 37 °C with shaking, stood still every 10 min and the absorbance was read at 600 nm.

### 3.6. ADME Properties

The absorption, distribution, metabolism, and excretion (ADME) properties of fourteen compounds were calculated by ACD/Percepta.

### 3.7. Pharmacophore Requirements

The GALAHAD module of Sybyl-X 2.0 was used to generate pharmacophore. Fourteen ocotillol-type derivatives that we had synthesised were selected with good activity against *B. sub*, including the compounds that at the C-3 position had free amino and carboxy group. All the structures are attached in Appendix B
Table A1 [16,17,18,23]. The final pharmacophore models were achieved with follow operations, including a population size value of 20, a maximum generation value of 100 and the value of molecular required hitting was 8.

## 4. Conclusions

In this study, based on the previous data and primary structure-activity relationship, a series of ocotillol-type derivatives were synthesized and tested for antibacterial activities against the gram-positive bacteria (*S. aureus*, *B. sub*) and gram-negative bacteria (*E. coli*, *A. baum*). It was discovered that these derivatives with free hydroxyl at C-3 have poor activity. However, the activities of derivatives substituted at C-3 with amino acid could enhanced activity against gram-positive bacteria and the antibacterial spectrum will expand when two free aminos were substituted at C-3; compounds **25** and **26** were such representatives. In addition, ADME property prediction showed that those derivatives with good activities also had fine ADME properties. Therefore, further research on this series of compounds is necessary. The results of the pharmacophores provided guidance for our next structural modification and transformation. Based on current research, we will focus on designing more derivatives as novel efficient broad spectrum antibacterial agents.

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
