# Peer review of "Design, Synthesis and Antibacterial Evaluation of 3-Substituted Ocotillol-Type Derivatives"

_molecules, 2018, doi:10.3390/molecules23123320_

Round 1
Reviewer 1 Report
I believe that antibiotic resistance is a very serious problem and needs a lot of attention from health and political authorities.
The authors propose triterpenes to solve the problem of antibiotic resistance. I think that is a great approach that needs to be investigated more and more.
The work is well written and I have few observations to make, in particular, the authors should add a reference to line 45, "Bonini SA et al. Cannabis sativa: A comprehensive ethnopharmacological review of a medicinal plant. J Ethnopharmacol. 2018 "
After these additions, the work can be published.
Author Response
Point 1: The authors should add a reference to line 45, "Bonini SA et al. Cannabis sativa: A comprehensive ethnopharmacological review of a medicinal plant. J Ethnopharmacol. 2018 "
Response 1: Thanks for your suggestion. According to your suggestion, the relevant literatures "Cannabis sativa: A comprehensive ethnopharmacological review of a medicinal plant." have been read carefully and the results have been referenced.
Reviewer 2 Report
In this manuscript, the authors describe the synthesis and antibacterial activity of a new series of derivatives of Ocotillol, a triterpenoid saponin. The study is the continuation of previous work on former derivatives of the same saponin, some of them having been published for example in Molecules in 2017.
The rationale for the study is sound and relies on former results obtained by the same consortium of research teams, which appears to have good experience in similar collaborative works. The synthetic scheme is well described and procedures seem to be sound and reliable. Nevertheless, reference 15 should be clearly noted in section 2.1 to precise how compounds 3 and 4 are obtained and separated. The new compounds are characterised by appropriate spectroscopic data. The anti-microbial activities of the compounds are interesting, some of them being quite active on a methicillin resistant Staphylococcus aureus strain.
Specific comments:
- The english wording needs much improvement all over the manuscript with the dramatic consequence, in several cases, of difficulty to understand the authors statements.
- Several compounds listed in annex A1 do not appear in the document such as compounds 2a to 2e, 3a, 3b
- Chemical structures in Figures 1 and Schemes 1 and 2 are too small to be easily differenciated and need to be enlarged.
- The authors state they have determined the ADME properties of 20 (R) compounds (line 146) but only 7 are given in Table 2. This is not clear since only 12 (R) derivatives are described in the manuscript. Furthermore the choice of the 7 compounds in Table 2 might be explained.
As a conclusion, the manuscript needs revision before acceptance.
Author Response
Point 1: The study is the continuation of previous work on former derivatives of the same saponin, some of them having been published for example in Molecules in 2017.
Response 1: Thanks for your suggestion. As you said, we had published a series of ocotillol-type saponin derivatives on Molecules in 2017. However, Scheme 2 shown in line 94 was synthetic route of our aim compounds 7-26 and all of the twenty compounds we synthesized in this manuscript are novel ocotillol-type derivatives.
Point 2: Reference 15 should be clearly noted in section 2.1 to precise how compounds 3 and 4 are obtained and separated.
Response 2: Thanks for your suggestion. According to your suggestion, the synthesis and separate methods of compounds 3 and 4 have been suppled in line 201-206, which make the experimental methods more complete.
Point 3: The English wording needs much improvement all over the manuscript with the dramatic consequence, in several cases, of difficulty to understand the authors statements.
Response 3: Thanks for your suggestion. According to your suggestion, the languages and expressions in the manuscript have been revised, which make the descriptions more rigorous and specification. The modifications in line 27-29, 60-62, 106-107, 108-109, 140-141, 164 and 166-167 marked in red.
Point 4: Several compounds listed in annex A1 do not appear in the document such as compounds 2a to 2e, 3a, 3b.
Response 4: Thanks for your suggestion. To establish a persuasive pharmacophore model, fourteen ocotillol-type derivatives we had synthesized with good activity against B.sub were selected. Six of the fourteen derivatives were from this document and the rests were from our previous articles. The references have attached in line 413-414
Point 5: Chemical structures in Figures 1 and Schemes 1 and 2 are too small to be easily differentiated and need to be enlarged.
Response 5: Thanks for your suggestion. According to your suggestion, the Figures1 and Schemes 1 and 2 have been revised carefully by enlarge the font and compact the space, which make it clearly to the readers.
Point 6: This is not clear since only 12 (R) derivatives are described in the manuscript. Furthermore, the choice of the 7 compounds in Table 2 might be explained.
Response 6: Thanks for your suggestion. In this part, seven compounds of 24(R)-configuration were selected in order to explore the relationship between amino substitution and the ADME properties. According to your suggestion, the discussion of ADME properties has been revised holistically in line 149-157 and 408-409. The predicted data of seven 24(S)-configuration compounds were added, which make this part of discussion more convincing.